# Dialogue Management and Language Generation for a Robust Conversational Virtual Coach: Validation and User Study

**DOI:** 10.3390/s23031423

**Published:** 2023-01-27

**Authors:** Alain Vázquez, Asier López Zorrilla, Javier Mikel Olaso, María Inés Torres

**Affiliations:** Speech Interactive Research Group, Universidad del País Vasco UPV/EHU, 48940 Leioa, Spain

**Keywords:** human–machine interaction, virtual coaching, spoken dialogue systems, natural language generation, artificial intelligence, user study

## Abstract

Designing human–machine interactive systems requires cooperation between different disciplines is required. In this work, we present a Dialogue Manager and a Language Generator that are the core modules of a Voice-based Spoken Dialogue System (SDS) capable of carrying out challenging, long and complex coaching conversations. We also develop an efficient integration procedure of the whole system that will act as an intelligent and robust Virtual Coach. The coaching task significantly differs from the classical applications of SDSs, resulting in a much higher degree of complexity and difficulty. The Virtual Coach has been successfully tested and validated in a user study with independent elderly, in three different countries with three different languages and cultures: Spain, France and Norway.

## 1. Introduction

The design and development of a human–machine interactive system involves solving difficult problems, for which the cooperation of different disciplines is required. First, the objectives of the interaction, the specific target community and the desired communication style must be defined, and then decide the technology and methods to be developed. The work presented in this paper was developed as part of the European EMPATHIC [1] project, which was aimed to research, innovate, explore and validate new interaction paradigms and platforms for future generations of personalised Virtual Coaches (VCs) to improve independent healthy-life-years of the elderly. As a result, a VC was developed under the guidance of Health and Coach professionals to provide it with coaching strategies and communication language [2]. On the one hand, the characteristics of the target community, i.e., independent elderly living in three very different countries, Spain, France and Norway, had to be considered. On the other hand, the interactions between the target population and the VC required a high degree of robustness because the VC was expected to carry out long and complex conversational coaching sessions.

The architecture of this VC can be considered an extension of typical Spoken Dialogue Systems (SDSs), with additional modules and capabilities. However, in this work, we focus on the Dialog Manager (DM) and Natural Language Generation (NLG) of EMPATHIC, the core modules of the VC, which are in charge of taking decisions and defining the communication style. The joint work of these two modules is capable of managing automatic, multilingual, complex and long coaching sessions aimed to enhance well-being by improving nutritional habits. This task significantly differs from the classical applications of SDSs, which typically consist of providing the users with information about timetables or facilitating the booking of services, resulting in a much higher degree of difficulty. Furthermore, we also describe the integration procedure of the whole system and its validation with real end-users who also provided measures of human acceptance of developed coaching dialogues, generated language and robustness. This VC is hardly comparable to any previously published system.

To develop the aforementioned modules, we leverage different Artificial Intelligence tools, mainly from the Natural Language Processing (NLP) and Planning fields. On the one hand, the DM represents the dialogue, designed under the guidance of professional coaches, as a tree of intelligent agents. Each agent is in charge of performing specific actions or subtasks within the dialogue. On the other hand, the NLG combines machine learning methodologies with classical NLG planning methods to produce accurate and correct sentences in the three target languages and in a style defined by health professionals.

As the main contribution, we validate our design by carrying out a study of the behaviour of the VC with end-users, in the three target languages and cultures, namely Spain, France and Norway. In order to validate it, first we evaluate its performance in terms of task completion, dialogue flow or number of repetition requests of the coach’s turns. We also compare it with a Wizard of Oz (WoZ) system in very similar conditions. The WoZ system is a seemingly autonomous system, but it is actually controlled by a human expert. As a result, we can analyse the differences between professional and automatic coaching, which is important to analyse to what extent the behaviour of the automatic system is similar to a professional coach. Finally, we briefly contrast the performance of our VC with an end-to-end dialogue model trained on the WoZ dialogues. Such comparison supports our approach for the core modules of the system against an end-to-end dialogue engine.

In this framework, the two contributions of this paper are as follows:On the technical side, we propose a closed-domain planning-based DM to develop complex coaching dialogues. We also present a hybrid NLG based on templates, augmented with a Transformer NN postprocessing and with a part-of-speech tagger based on word vector representations, to implement the coaching language of the VC in three languages under the guidance of health professionals. To put these contributions into effect, an elaborated Dialogue Act (DA) taxonomy that routes the communication between the DM and the NLG has been developed, and also an efficient integration procedure for the assembly of the VC.Secondly, we carry out a study to test, analyse and validate the VC in the target community, elderly independent people, in three different countries with three different languages and cultures: Spain, France and Norway.

The rest of the paper is organised as follows: Section 2 summarises the main related work. Section 3 shows a general description of the modules that make up the VC that executes the coaching approach selected for this work. Then, Section 4 explains how these coaching dialogues are represented and managed by the DM. Section 5 illustrates the DAs proposed as a communication procedure between DM and NLG, and Section 6 describes the NLG proposed to generate the coaching language of the VC. Next, Section 7 details the procedure followed to integrate all the modules and technologies and thus compound the VC. Finally, Section 8 presents the evaluation of the described methodologies as well as the behaviour of the VC during coaching sessions aimed at improving nutritional habits. We end with some conclusions in Section 9.

## 2. Related Work

### 2.1. Dialogue Management and Natural Language Generation

Unlike other areas of AI, Dialogue Management lacks any commonly used framework or tools to build data-driven baselines for research works or stable and production-ready prototypes. There have been many efforts to develop and establish different kinds of technologies to this end, but none has ever become the go-to approach to build DMs. The situation regarding NLGs is similar too, but this has not been the case in other areas. For example, acoustic models for Automatic Speech Recognition have typically relied on Gaussian Mixture Models and Hidden Markov Models for many years, and only in the last decade they have been augmented with Neural Networks (NNs) [3]. Similarly, Convolutional NNs have been a reliable tool in Image Processing, mainly since the release of ImageNet [4], which enabled the pretraining of large general-purpose models which could then be fine-tuned from many applications, serving as a powerful baseline and starting point for many image-related projects. Another example can be found in Machine Translation, where sequence-to-sequence [5,6,7] models marked a watershed in the history of the research area.

However, such turning points have not occurred in Dialogue Management. One problem is that every DM is different because the applications, task, language and available resources, such as data, are different in almost every project. Another problem is that human, natural dialogues are extremely complex and difficult to model, which has typically caused every proposed methodology to be either too hard and laborious to implement, or to produce many errors and not to be usable in real-life situations. Stochastic approaches based on Markov Decision Process [8,9] were the first statistical proposals to deal with Spoken Dialogue Systems. Then, some approaches based on Stochastic Finite State Transducers or bi-automata [10,11] have also been developed. Nevertheless, one of the main efforts to establish a common framework to develop dialogue models were POMDP-based DMs [12,13]. However, their complexity, difficult applicability to real-world tasks, and the rise in popularity of NNs slowly gave way to deep learning-based approaches to dialogue management. Deep learning-based open-domain chatbots, such as [14,15] or [16], surely suppose one of the biggest recent breakthroughs in dialogue management. However, the amount of data needed to build such models and the little or no control of their behaviour once they have been built make them infeasible to implement for many practical applications. Even though deep-learning-based chatbots can be built with rather a few amount of domain-specific data for goal-oriented tasks [17,18,19], the problem of the lack of control of the model still remains. This often causes a lack of robustness which can hardly be avoided, and that is the reason why health-related data-driven DMs are extremely scarce in the literature. Among such systems, we can find [20,21], but these have not been tested with potential end-users. On the other hand, even though there are some other works describing DMs for tasks comparable to ours, the vast majority of them employ agenda, plan or rule-based designs (see Section 2.2). Similarly, and due to the drawbacks of data-driven DMs, the design proposed in this work is based on a structured plan of the dialogues.

Regarding NLGs, a large variety of novel approaches can also be found in the literature. Nonetheless, these have not replaced the more classical technologies, and the selection of which one to use is still very dependent on the task. For example, among the 63 different NLGs that took part in the end-to-end NLG Challenge [22], a wide range of technologies can be found: from novel NN architectures or other data-driven systems to more classical rule or template-based generators. Even though sequence-to-sequence NN systems were the winners of that challenge, in some cases, classical approaches obtained similar or better results in terms of human evaluation. Additionally, Refs. [23,24] highlight the difficulties that data-driven machine learning-based NLG systems have for real and, more specifically, medical applications. In general, these systems have problems generalising with complex inputs and are not reliable due to the lack of control of the output. To avoid these drawbacks, whilst [24] employ conceptual graphs that allow for including high-level knowledge, we propose a methodology to obtain the best of both worlds. We build an NLG that combines the classical template-based approach with transformer-based neural language models (LM). The template-based approach provides us with flexibility and control, whereas the neural LM will help to post-process and improve the quality of the generated sentences.

### 2.2. Virtual Coaches

Once we have provided a high-level view of the methodologies usually employed in dialogue management and language generation, let us now present some works describing dialogue engines, i.e., the joint work of DM and NLG, built to carry out coaching tasks. First of all, we should underline that the implemented coaching methodologies and their area of application greatly vary among the academic works and products in the market. Regarding the coaching strategy, we would like to clearly differentiate two kinds: goal trackers and coaching via motivational conversations. Goal trackers aim to track and visualise the progress of the users towards their—often simple in terms of definition—goal. An example are applications that monitor the amount of daily physical exercise completed by the user, or the number of steps. On the other hand, coaching via motivational conversations is radically different. The coach should be able to provoke reflection through dialogues and thus assist the user in finding a goal as well as the next steps needed to reach it. In this work, we focus on a coaching methodology based on conversations.

Some examples of goal trackers that also include some kind of dialogue engine are [25,26,27]. The work presented in [25] describes CoachAI, a task scheduler and tracker that promotes physical activity. Its DM is based on a structured finite state machine, which guides the user through a series of steps to achieve their daily goal. The system in [26] also aims at promoting physical activity, in particular regular aerobic exercises. The users first set their weekly goals; then, the coach keeps track of them, schedules exercises and offers future objectives depending on the users’ progress. Finally, Ref. [28] presents an embodied conversational agent to help students manage their study stress. They focus on understanding the users’ goal and provide explanations about one of four predefined behaviour change procedures. However, the dialogue engines of these works are not designed to support long or complex interactions.

Closer to our domain of interest, recent reviews [29,30,31] show some conversational agents developing well-being tasks. These agents tackle depression, anxiety, autism or chronic diseases, among other health issues. However, the systems presented in these reviews (Woebot [32], Tess [33], MMD agent [34], Avachat [35] or JADE [36]) do not base their coaching methodology on conversations. Instead, the interaction just directs the coachee to audiovisual elements such as video links. In the same way, Ref. [31] develops chatbots dealing with mental health and highlight that the development of health conversational agents is still in its infancy. The SDS proposed in this work, which is capable of carrying out much longer dialogues, contributes to the advance in this area of research.

Nevertheless, there are some virtual agents that carry out a coaching or well-being task based on a conversational approach. In this case, a variety of dialogue technologies can be found, as well as different degrees of freedom in which users can express themselves. It should be noted that some of these conversational agents implement just very closed dialogues in which user responses are limited to a list of multiple choices. These systems employ a rule-based DM and an NLG based on templates or on predefined statements. Some examples are the prototype of the VaSelfCare project [37], the CADLAC system [38] and the virtual therapist presented in [39]. Nonetheless, these works also show some similarities with our proposal. In fact, the objectives of the VaSelfCare project are focused on the nutrition, physical activity and medication habits of the users, but targeting people with diabetes. On the other hand, CADLAC exemplifies the difficulty to integrate NN approaches in real applications. Its prototype, presented in [40], was designed to include a sequence-to-sequence machine translation NLG model, but later results presented in [38] were obtained with a rule-based NLG. Conversely, our work integrates a DM that accepts unconstrained natural language inputs, which entails a higher complexity, and an NLG that incorporates a NN technology.

Closer to our proposal, some works allow free language input. Wysa chatbot [41,42] works with free natural language or just a multiple-choice input, depending on the dialogue state. As for the output, unrestricted responses in complex dialogues can be found in EVA, MIRA and AMANDA [43,44,45], which are virtual medical assistants developed within the RASA (https://rasa.com, accessed on 20 January 2023) framework. In the same way, BLISS [46] is a conversational agent aimed to discover what makes people healthy, based on the Flipper dialogue engine [47]. Another approach is proposed in [48], where a conversational agent aimed to assist people with language and communication problems is developed with AIML technology. After classifying the input, it selects the outputs from a list of templates.

Finally, we would like to underline that the EMPATHIC VC is multilingual and multicultural. Whereas the majority of the SDS and conversational systems mentioned in this section were deployed in only one language, usually English, the VC described in this paper is running in Spanish, French and Norwegian. These languages have far fewer linguistic resources for the development of AI models, notably Norwegian, which makes the task much more difficult. As far as multiculturalism is concerned, the target community includes users from three countries: Spain, France and Norway.

## 3. Virtual Coach Overview

This section provides a comprehensive view of the EMPATHIC VC, in order to present and contextualize the VC system that we validate later in our study. First, we briefly describe the modules integrating the system in Section 3.1. Then, Section 3.2 describes the coaching methodology selected to implement the dialogue management and communication style/language of the VC.

### 3.1. Main Components of the Virtual Coach

The VC consists of a multimodal automatic dialogue system whose users can interact with through any device that has a web browser (PCs, Tablets or Smartphones, for example). The main components of the VC are shown in Figure 1, highlighting the modules we discuss in this work in green.

All the modules have been developed by EMPATHIC partners as part of the project. In this work, we only detail the proposed DM and the NLG modules in Section 4 and Section 6 as they are the core modules to implement the behaviour of the VC. In addition, Section 5 presents the DA as the communication protocol between the DM and the NLG. Descriptions of other modules can be found in other works: NLU [49,50,51], ASR [52], biometry [53] and emotion analysis [50,54,55,56,57,58].

### 3.2. Coaching Approach

According to the Spanish Coaching Association [59], professional *coaching* is a personalised training process aimed to cover the gap between the current personal situation of the coachee and their desired situation. For a proper strategy implementation, the coach has to understand what the client is willing to do in order to promote a behavioural change and acquire a commitment to it. Consequently, the desired evolution is defined by the coachee.

In the EMPATHIC project, the GROW model [60] was the chosen strategy implemented by the VC. It provides a simple and effective way of structuring coaching conversations in a sequence of stages (Goal, Reality, Options and Will), which is not completely linear since a stage can always be revisited if needed. Moreover, such a strategy is mostly guided by questions. Hence, Figure 2 shows the sequential structure of the model and the type of questions the VC asks at each stage. The process, adapted from [61], is defined as follows:

Goal Phase.The VC implements Goal Set Questions (GSQ) and Motivational Questions (MQ) to identify and clarify the user’s goals and commitment to them.Reality Phase. The VC assesses the current situation via Reality Questions (RQ) and provides an understanding of internal obstacles with Obstacle Questions (OQ).Options Phase. The VC implements a variety of Option Generation Questions (OGQ) to identify alternatives to achieve the goal defined by the user. It also uses Obstacle Questions (OQ), in this case, to assess the difficulties of these potential action plans.Will Phase. The VC implements a series of Plan Action Questions (PAQ) aimed at defining an action plan to reach the previously set goal.

Additionally, the VC can also employ Follow-up Questions (FQ) to check the progress after a coaching session and Warning Questions (WQ) to assess risk situations regarding the health status of the user.

We have been assisted by a professional Coach for the design of the DM strategy and the style of the language produced by the NLG. She developed a series of dialogues simulating real coaching interactions for four different scenarios, which we tried to reproduce automatically: (a) nutrition, (b) physical activity, (c) leisure and (d) family and social relationships [62,63,64]. Figure 3 shows an example of such dialogues developed for the nutrition scenario.

## 4. Dialogue Management to Perform Coaching Strategies

In this section, we first describe the proposed DM in Section 4.1. Then, Section 4.2 reports the task specification tree proposed to implement the introductory dialogue and the coaching strategies based on GROW model for a nutrition scenario.

### 4.1. Dialogue Manager

For the management of the EMPATHIC dialogues, we have grounded our design on a planning-based DM [65] that we have squeezed the most out to suit the needs of the project. This DM achieves task independence by decoupling the domain-specific tasks and the domain-independent control and execution mechanisms. It is flexible and scalable enough to address different domains needing different planning and communication skills. However, the tasks previously addressed are quite simple, mainly consisting of information-access [11,66,67,68] or scheduling [69], which strongly differ from the coaching strategies that the VC has to manage in this work. Nevertheless, the main advantages of using this DM are its suitability for the development of SDS in complex task domains and its capability to allow users a fairly unconstrained natural language. In this work, we have taken advantage of these features to go beyond what has been achieved so far, and manage complex dialogues that the coaching strategies need.

This DM develops a management structure based on distributed software agents that first specify the dialogue task at the design level and then execute the dialogue flow at the running time, as follows:Dialogue task specification. It follows a hierarchical plan that is defined by a tree of dialogue agents, where each agent is responsible for managing a specific subtask. Two different kinds of agents can be found in the tree:-Internal agents or non-terminal nodes, which are represented as blue nodes in Figure 4, are used to encapsulate subsections of the dialogues and control the execution of their children agents;-Terminal nodes, mostly represented in red in the figure, are responsible for implementing precise actions. In this way, Inform nodes produce an output, Request nodes ask for some information from the user, and Expect nodes continuously listen for some information without requesting it. The green nodes in the figure are Execute nodes connected to other modules of the SDS.Dialogue management. The DM executes a given dialogue task specification tree that is traversed in Depth First order. However, this order can be altered under specific preconditions, triggers or success/failure criteria of the internal agents. The DM uses two structures to traverse the tree:-A stack that assists the Depth First search by storing the dialogue flow (Figure 5);-A dashboard that stores information provided by the user or by external sources that is useful to keep the consistency of the dialogue.

As an example, Figure 4 partially shows the dialogue task specification tree of the introductory dialogue, and Figure 5 shows how the DM traverses it. Both non-terminal and terminal nodes are inserted in the top of the stack following the Depth First order and then popped from the top of the stack once its execution has finished. Non-terminal, such as Introduction or IsFirstUse, nodes require all the children to be executed before being popped from the stack. However, the execution of terminal nodes (red and green in the figures) consists of individual distinctive actions that are executed and immediately popped from the stack.

In addition, trigger conditions of the internal agents might make the DM push a particular agent to the top of the stack and thus be executed immediately. As proof of concept for this capability, we added an agent that is able to provide users with culinary recipes. When this agent detects that a particular food has been mentioned repeatedly during the conversation, this agent is instantly placed at the top of the stack, and the DM provides the user with a recipe related to that food.

Finally, we integrated an emotion listener that was implemented as a dialogue agent receiving information from the module in charge of the Emotion Analysis in Figure 1. This agent is responsible for updating an internal variable storing the last emotion detected in the dashboard. Then, each time the VC has to generate a response the DM assigns an emotion to the voice of the virtual agent according to the user emotion identified [2,70].

### 4.2. Coaching Strategies for the Introductory Dialogue and the Nutrition Scenario

Figure 4 and Figure 6 show the task specification trees for the introductory dialogue and the nutrition scenario, respectively, which can be considered the two parts of the coaching session. The introductory dialogue deals with user-friendly topics aimed to obtain basic information about the users, and also provides some context about coaching and the EMPATHIC project. The nutrition scenario is the coaching session, based on the GROW model presented in Section 3.2. Both dialogues were designed based on two references: the documents provided by the professional coach mentioned in Section 3 and a corpus of dialogues between elderly people and a simulated VC acquired through a Wizard of Oz technique [71].

As shown in Figure 4, the introductory dialogue is different depending on whether it is the first time the user talks to the system or not. If it is the first time, the system creates a biometry profile, welcomes the user, asks for their name, and briefly talks about the weather and coaching. Otherwise, the system authenticates the user and, after greeting them, it jumps to the GROW session about nutrition.

Figure 6 shows that the sub-agents of the nutrition agent are highly connected to the GROW model. First, the GROW phases are preceded by an introductory nutrition dialogue. This first part of the nutrition session is designed to look for potential inappropriate habits by examining users’ intake routines in their daily meals. Regarding the GROW phases, the Goal (G) agent is aimed at obtaining a nutrition goal from the user, which can be one detected by the coach in the nutrition introduction, one proposed by the user or one selected from a predefined tree of goals. Next, we insert a Motivation (M) phase within the GROW phases with the aim of exploring the motivations leading the users to change their nutritional habits, if any. In addition, the remaining phases were designed to follow the directives of the GROW: Reality (R) agent to find out what are the main problems users have to achieve the goal, and how far they are from it, Options (O) agent to explore the users’ options to achieve the objective previously defined, and Will (W) to obtain a specific action plan that users should execute in order to achieve the identified goals. Finally, and before closing the session, the system performs a short summary with the main points and decisions taken, in order to verify the users’ commitment to the goals and the next steps to be taken.

## 5. Design of the Dialogue Act

In Spoken Dialogue Systems, the DA is in charge of representing the meaning of the system responses. It acts as a communication element between the DM and the NLG since it is the output of the DM and the input of the NLG.

Although the DA can be designed in different formats, the typical design consists of a principal component, *DA intent* that defines the communicative intent of the sentence to be generated and a set of attributes that stands for the semantic content of the sentence [22]. Our DA intent consists of a hierarchical label, as we explain in Section 5.1. In addition, our *attributes* not only define the semantic content but also deal with the grammatical correctness.

Table 1 shows a segment of a real interaction between a user and the EMPATHIC VC. For the user turns, the textual transcription and the detected entities are shown. Regarding the coach turns, which are our focus, we show the DAs (column 2) and the generated sentences (column 3). Throughout this section, we illustrate the design of the DA with the examples in the table. First, we define the DA intent in Section 5.1, and later the attributes are shown in Section 5.2.

### 5.1. DA Intent

The *DA intent* is an element present in all DAs. It is composed of a label and a sublabel that provides a double functionality. The label informs about the section of the dialogue in which the sentence is going to appear, and the sublabel complements it by providing semantic or communicative intent information.

Regarding the label, there are two possible contexts in our dialogues described in Section 4: a coaching context and an introductory part. In addition, there are sentences that can appear in both contexts. Consequently, we split ten possible labels into three blocks:**GROW**. The eight labels of this block are the eight question types of the GROW model shown in Figure 2. Thus, RQ, PAQ and GSQ only appear in the coaching context (*Mid-conversation* and *End of the conversation* in Table 1).**Introduction**. This block includes a unique label denoted as Int and covers all the sentences that only could be found in an introductory dialogue, such as “What is your name?” or “My name is Natalie. Nice to meet you”. Thus, in Table 1, we find this kind of sentence labelled as Int only in the *Start of the conversation*.**Task independent**. This block includes a unique label denoted as Gen. It encloses all the sentences that can appear in any context-like greetings, thanking or backchannels, among others. Table 1 shows that domain-independent sentences labelled as Gen may appear in any part of the dialogue.

These ten labels are complemented with more than 100 sublabels although each label only accepts a set of sublabels. In the domain-specific blocks (GROW and Introduction), the sublabels inform about the semantic content of the sentence to be generated. Conversely, domain-independent sublabels determine the communicative intention of the sentence.

In domain-specific examples of Table 1, we find, for example, Int&what_name() or RQ&curr_sit(<<topic>>=breakfast). The first example appears in the introductory part as its label indicates. In addition, the semantic content of the sentence *“What is your name?"* is defined by the sublabel what_name. Similarly, curr_sit (contraction of *Current situation*) determines that the semantic content is related to the current situation of the user, while RQ contextualises the sentence in the Reality phase of the GROW model.

For the Task independent block, the sublabels were defined based on two well-known sets of labels, the SWBD-DAMSL corpus and the DIT++ taxonomy [72,73]. They are communicative intents such as Hello or Inform that appear appended to the label Gen in the table and can be found in any context of the conversation.

### 5.2. Attributes

The example Int&what_name() shows that sometimes label&sublabel are specific enough to define the sentence to be generated. However, for example, the second DA of the first turn in Table 1 needs a complement for Gen&Inform to know about what the coach has to inform. Such specific information is provided through the attributes. In the example, <<biometry>>=no_user determines that the information to convey is that there is not a profile created for this user.

In this work, we consider two types of attributes: contextual and replacement attributes. Both of them send contextual information that helps to generate the sentence, but in the case of the replacement attributes, their value has to be explicitly included in the sentence. In the previous example, Gen&Inform(<<biometry>>=no_user) that generated *“I need you to sit in front of the camera..."* does not include the value no_user in the sentence. However, the sentence generated by the DA Gen&Open_topic(<topic>=breakfast) includes the replacement attribute: *“Let’s focus on your breakfast."*. To distinguish both types, contextual attributes are represented as <<attribute>> in Table 1, whereas the replacement ones are depicted as <attribute>.

Table 2 shows the importance of the gender attribute in the generation for Spanish and French. Thus, the genders of the user and coach are sent as part of a special kind of attribute, named *Conversation profile attributes* that also include the language, the user and coach names or the polarity, among others. They are not shown in Table 1 because they are a set of high-level attributes that are not only shared between DM and NLG, but also with the rest of the modules.

Additionally, there are also language-independent replacement attributes. These attributes are generated by the DM and have to be transformed into the target language by the NLG. For instance, in the last turn of Table 1, the DA contains the attribute value <action>=eat_regular, which is generated as *“improve your regularity"*. Table 3 shows that, for each value, there are a number of different possible generations (referred to as *versions* in the table) to increase the variety in the responses.

The complexity of our multilingual task and dialogues is reflected in the design of the DAs. First, we find a great variety of DA intents. Moreover, there are attributes of very different natures. Consequently, the NLG developed for this work has to be adapted to this complexity, which involves processing and integrating all the information communicated through the DAs.

## 6. GROWsetta: Natural Language Generation for Coaching

The language generator integrated into the final prototype is a multilingual template-based system developed from scratch for this project: GROWsetta. Appendix A shows that GROWsetta has been selected in a comparison with an end-to-end approach because it is faster and more controllable, while generating sentences of similar quality.

Template-based systems map conceptual (non-linguistic) representations to templates. Such templates are linguistic structures with possible slots that are filled to obtain well-formed sentences. This procedure is adopted by GROWsetta, but we include additional steps due to the complexity of the coaching task. Thus, GROWsetta converts the DAs presented in Section 5 to natural language responses in a process that can be split into five steps.

### 6.1. Step One: Augmented Part-of-Speech Tagger Task over the NLU Entities

First of all, GROWsetta analyses the entities detected by the NLU and sent to the NLG as DA attributes. This is needed because the entities do not follow any specific format in their values. Hence, we need to consider which kind of word the entity is, as well as its grammatical features to ensure correctness when we introduce the entity in the sentence.

Three analyses are integrated into our pipeline, as depicted in Figure 7. The first step of our proposal is a morphological analysis carried out by a tagger (included in the NLTK toolkit (https://www.nltk.org, accessed on 20 January 2023)) to infer the linguistic structure of the attribute (noun, determiner + noun, verb, …). Then, if the linguistic structure of the attribute value is a noun, we determine its gender, number and countability with a self-deployed classifier. Finally, an additional and non-conventional analysis for this task is carried out, if the attribute is <food>. It is used to determine whether the value is a meal or a drink.

The noun and food analysis are performed by a set of classifiers: gender, number and countability for nouns and one for the food type. Each classifier works with two bags of words that represent each category, as shown in Figure 8. The attribute value is assigned to the category whose words are closer to it.

All in all, there are some kinds of entities that cannot be processed in this step. For instance, it is not straightforward to determine whether a date recognised by the NLU is past, present or future. We tackle this issue in Step 5.

### 6.2. Step Two: Transforming Language-Independent Values

Next, if there are language-independent replacement values, GROWsetta has to transform them into language-specific expressions to include them in the gaps of the templates. For this purpose, GROWsetta has multiple predefined transformations for each possible value of these attributes, as it has been explained in Section 5.2 and shown in Table 3.

### 6.3. Step Three: Template Selection

In this step, given the DA intent and based on the restrictions established by the attributes, a set of grammatically and semantically correct templates are selected. The replacement attributes force the template to have slots for their values, while the contextual ones select only the templates that fulfil their requirements in terms of semantic and grammatical content. Moreover, the analysis of the entities of the first step also provides grammatical restrictions for this selection.

### 6.4. Step Four: From Templates to Sentences

In this step, the NLG replaces the slots in the templates previously selected (step 4) with the values of the attributes already analysed, adapted and transformed (steps 2 and 3). Thus, well-formed sentences are generated.

### 6.5. Step Five: Selecting the Best Sentence (Transformer Postprocessing)

At this point, GROWsetta has generated a number of candidate outputs. Depending on the attributes included in the DA input, we cannot ensure the grammatical correctness of the responses for all the candidates. In those cases, we carry out a neural language model postprocessing to select a correct sentence. For example, Table 4 shows two situations where only one of the two candidates is correct and such postprocessing would be necessary.

We propose to use a LM of the target language to select a correct option among the set of candidates. The LM is a statistical model that provides a probability to a sequence of words. Thus, GROWsetta selects the candidate for which the LM estimates the highest probability. Among the available options [74,75], we decided to use the Generative Pretrained Transformer 2 (GPT-2) neural LM architecture, which has proved to be successful in many natural language processing tasks [76]. We train these networks from scratch for the three target languages. For this purpose, the Spanish, French and Norwegian versions of Wikipedia and OpenSubtitles [77] corpora were selected due to their availability for the three languages and variety in topics. A fraction of the Norwegian version of the OSCAR text corpus [78] was also included since the amount of data for Norwegian was much lower than for Spanish and French. Table 5 shows some statistics of our training data for each language.

We also tested a simpler approach such as the well-known statistical N-grams model. However, Section 8.5.1 shows that its performance was clearly worse than the one obtained by the transformer-based LMs.

## 7. Integration

The system was built using a client–server paradigm in which users interact through a client application that captures their inputs and submits them to the server. The client application makes use of the native capabilities of the users’ devices (phone, tablet or PC) to relay audio and video to the application server. Thus, it uses the device’s own camera and microphone to sense the users and also displays a talking visual agent. This system is hosted in three servers located in London, one for each language. Each server contains three instances of the system, so that three users can interact with it at the same time in each language.

The system architecture consists of several components that can be classified into four broad categories: infrastructure components (Section 7.1), application components (Section 7.2), websites or services (Section 7.3) and external services (Section 7.4). A schema of the architecture is shown in Figure 9.

### 7.1. Infrastructure Components

Infrastructure components are elements that do not provide specific functionalities of the dialogue systems but act as supporting elements for building them. In the system developed, these components are:Software Containers Framework. In our system, all the components are running within a Docker container [79] (orange in Figure 9), except for the components used as external services (ASR and TTS). Docker is a software platform that enables developers to create, test, and deploy applications within containers. A container is a packaging format that can encapsulate applications along with all their dependencies.Events and Message Broker. Roughly speaking, a message broker can be defined as a message transfer agent between different applications. For this system, Apache ActiveMQ [80] has been used because it is Open Source and allows communication between applications written in various programming/scripting languages.Web Applications/Sites Server. Apache Tomcat (http://tomcat.apache.org/, accessed on 20 January 2023), represented in light yellow in Figure 9, is the web server of the system. It allows hosting the web page to access the system. Indeed, it can implement complex applications and offer them as web services. In our case, such services are mainly aimed at the exchange of multimedia elements (user audio and video, VC audio) between user devices and the system server.

### 7.2. Application Components

We consider in this category the components that make up the system as such, and they are the ones mentioned in Section 3. They are represented as brown boxes in Figure 9. These elements communicate with each other by sending text messages through queues of the Message Broker described in the previous section (red arrows in Figure 9).

### 7.3. Web Sites and Services

In this section, we describe the components (dark blue components in Figure 9) that are part of what we call the input–output layer of the system. It is composed of web pages that implement the user interface and web services to exchange data between the user interface and the server machine that hosts the system.

User Interface. The user interface consists of two web pages (Figure 10). Figure 10a shows the page to choose the avatar of the VC the users want to interact with and asks users to provide some basic data (user name, gender). Then, users interact with the system through the second page, which consists of an avatar and some control buttons (Figure 10b). On the one hand, the avatar is a 3D character designed with CrazyTalk (https://www.reallusion.com/crazytalk/, accessed on 20 January 2023) and imported into Unity [81] with a set of body and facial animations. On the other hand, the control buttons let the users start or stop a session as well as record or write their turn. Finally, the user interface accesses the users’ cameras and microphones and sends the audio and video data to the server using the capabilities provided by browsers.Audio Interface. This service receives the audio data sent by the user interface. It has two main tasks. The first is to serve as an intermediary between the user interface, the speech recogniser service and the NLU. The other task consists of redirecting the audio data to the audio emotion detection component. In addition, it is responsible for storing the audio data with the transcriptions in the server for future use.Video Interface. The main objective of this service is to redirect video data to both the video emotion analysis and the biometric analysis components. In addition, it is also responsible for storing the videos of the sessions on the server for later use.Text to Speech Interface. The purpose of this service is to act as an intermediary between the NLG module, the text-to-speech service and the avatar. Whenever it receives a sentence to be uttered by the agent, this component sends it to the text-to-speech service, waits for the corresponding audio file, and then sends it to the avatar. It has to be noticed that this service also sends the associated text string as well as an associated animation name along with the audio file. This information is initially sent from the DM to the NLG, and from there to this component.Dialogue Manager Interface. In order to start and finish dialogue sessions the DM needs to receive some specific messages. This service is responsible for forwarding these messages from the user interface to the DM.

### 7.4. External Services

Our system also uses some external services (yellow components in Figure 9). From the modules presented in Section 3, the ASR and the TTS are external services. For ASR, we developed a configurable system that allows for choosing between the following providers: Google Speech to Text (https://cloud.google.com/speech-to-text, accessed on 20 January 2023), Intelligent Voice ASR [52] or ASR from the Acapela Group (https://www.acapela-group.com, accessed on 20 January 2023). The Acapela Group provided the TTS service too.

Finally, the system provides the users with weather forecasts or culinary recipes at some points in the dialogue. These data are extracted from resources available on the Internet.

## 8. Results

In this section, we present the results obtained in the study of the EMPATHIC VC developed in this work. The main purpose of these experiments was to validate and measure the performance of the EMPATHIC system as a whole, which also allows us to analyse the behaviour of DM and the NLG. We also compare these results with the Wizard of Oz (WoZ) data acquisition experiments carried out previously in EMPATHIC [71], as well as with a fully data-driven coaching chatbot developed with this WoZ data [21], whenever possible. If the reader is interested in seeing an example of the automatic GROW sessions guided by the proposed dialogue engine, we refer them to [70], where a video of a session in French subtitled in English is shown.

We first describe the profile of the participants in these experiments in Section 8.1. Then, Section 8.2 offers a first glimpse of the conversations carried out with the system, with some general statistics about dialogue and turn lengths. These statistics are compared with the EMPATHIC WoZ system and the data-driven coaching chatbot. Subsequently, we validate the implementation of the DM, showing in Section 8.3 that the test conversations follow the dialogue flow planned in Section 4. Afterwards, we further validate the DM showing in Section 8.4 how many of the GROW stages were completed during the tests. Later, Section 8.5 provides measures of the performance of the NLG in the VC, as well as a preliminary analysis of its LM postprocessing. We end with the presentation of questionnaires about the human acceptance of the system in general, and the DM and NLG in particular in Section 8.6. Again, these results are compared with the WoZ system.

### 8.1. Participants’ Profile

In total, 79 elderly participants took part in these tests: 31 in Spain, 22 in France and 26 in Norway. Table 6 shows participants’ profile, which was partly obtained through questionnaires filled out before the interaction with the system. From the table, the quality of life of the participants was measured on a 0 (very poor) to 100 (very good) scale via the WHOQOL-BREF questionnaire [82]. Their depression level was measured with the GDS questionnaire [83] in a scale from 0 (no depression) to 30 (great depression). Finally, the participants were asked about how easily they can manage PCs in a scale from 1 to 5 which has been converted to a score from 0 to 100.

The profile was exactly the one the EMPATHIC project aimed at: healthy elderly people with a rather great quality of life, which also leads to low depression scores in most cases. The participants were also comfortable using PCs, in general. Thus, our results can be extrapolated to human–machine interaction with this particular group of people.

### 8.2. General Dialogue Statistics

Let us now present some statistics of the dialogues carried out between the participants and the SDS. Figure 11a shows that the system was able to keep long conversations. The Spanish tests were carried out first, before the French or Norwegian tests. The Spanish system presented some misbehaviours that were corrected subsequently. Thus, French and Norwegian conversations were significantly longer than in Spanish because the French and Norwegian versions were more stable, and fewer dialogues had to end prematurely. However, an additional reason for the longer conversations in French and especially in Norwegian is the lower ASR performance. In these languages, users had to repeat some information more frequently until the system correctly understood it.

The distribution of the number of words per user turn is shown in Figure 11b, divided per language. This metric is very relevant because longer responses often correlate with higher user engagement [84]. Intuitively, if the users are comfortable talking to a system and they feel they are being understood correctly and responded to coherently, they are much more likely to be more talkative and provide more information. On the other hand, if the system is having trouble understanding what the user is saying and makes them repeat information frequently, chances are that they will answer with much fewer words so that the system understands them better.

The distribution shown in Figure 11b indicates that, even though the dialogues were quite long, not all the user turns were so. Many turns were made of very few words, but a few of them were really long. As aforementioned, this depends on the willingness of the user to interact with the system, but also on the system’s questions. In any case, longer responses were produced in Spanish, then in French, and the shortest were in Norwegian. Shorter responses in Norwegian were conditioned by worse performing ASR and NLU, caused by the lesser amount of language resources for those languages. On the other hand, the difference between Spanish and French might be due to Spanish being the mother tongue of the main developers, and thus the system was tested mostly in this language, leading to fewer understanding errors. Cultural aspects of the users could also be a potential explanation.

Table 7 compares the average turn length (see Figure 11b) with the EMPATHIC WoZ experiments [71] and data-driven chatbot [21]. The WoZ experiments and the VC tests were carried out in very similar experimental conditions: the systems looked similar, they were tested by elderly participants, and the conversations were spoken. However, it is important to mention that the chatbot was text-based and was tested by health-related professionals, not by elderly. Nonetheless, the results should still be comparable to some extent because the task was the same and the chatbot was trained with the WoZ data. Note that the design of the VC was also based on these data. In the table, the star symbol (*) indicates when a result is significantly different than its counterpart. More specifically, it means that *p*-value ≤ 0.05 using Welch’s *t*-test, which tests whether two populations have equal means, without assuming equal variances. We use such a statistical test and *p*-value threshold in all the comparisons in this chapter. In this case, we compared the VC with the WoZ system on the one hand, and the VC with the bot on the other hand.

The user engagement level was significantly lower in the tests with the automatic VC than in the WoZ trials, as expected. Even if the difference could be partly explained by the conversational style of the wizards (which may use more open questions than the automatic prototype), it still indicates that users are much more talkative when talking to humans (or, for that matter, to a WoZ system) than to automatic systems. Regarding the chatbot vs. VC comparison, the VC led to longer user turns than the chatbot, especially in Spanish and French. This highlights the current limitations of fully data-driven approaches and ratifies our choices for the DM and NLG methodologies.

Finally, we also report the average response time (and standard deviation) as a measure of the efficiency of the proposed VC. This metric refers to the time between the end of the user speech until the coach starts its turn, and is shown in Table 8. The differences between countries were largely dependent on the end-users’ connection quality and distance to the servers, located in London. In any way, and even though the system’s performance can be improved in this regard, we find it acceptable for a prototype (the latency can be observed in the EMPATHIC demo [70]), and there were no big concerns from the end-users either.

### 8.3. Dialogue Flow

Let us now show the dialogue flow the DM produced in the tests. Figure 12 shows this dialogue flow in the form of a directed graph. The system turns with similar DA’s semantics in the dialogues are grouped into different nodes. The arrows represent significant transitions, i.e., an arrow from a node A to a node B indicates that system turns grouped in node B have followed turns grouped in node A at least 10% of the times node A was visited. This is carried out to keep the graph clearer and more representative. The nodes are coloured according to the dialogue phase they belong to.

First of all, the sequential nature of the automatic GROW sessions can clearly be seen in Figure 12, which means that the implementation and the design of the DM were correct and that the system acted as expected. In the graph, the nodes of the same dialogue phase are clustered together, and they only precede nodes of the same phase or the next one. The main exceptions are premature endings of the session. Such premature endings happen when the user and the system do not successfully achieve the objectives of a particular phase. For example, the session might finish in the Options phase if the system is not able to understand the next steps the user proposes to achieve their goal, and, similarly, it occurs for the Goal or Will phase.

Cycles within the dialogue phase can also be recognised in Figure 12. On the one hand, self-loops (a node with a transition to itself) are due to the nature of the nodes of the graph: they are not system turns, but groups of them. For instance, the self-loop in *request user name* happens because, after the system asks the user name, it tries to confirm it in the next turn, but these two turns are gathered in the same node. The other loops that often appear in the graph are transitions from the nodes of a phase to the first of the corresponding phase. This occurs mostly in the Will phase (see the transitions from the nodes of the Will phase to *goal opening*), but also in most of the phases. Since the conversations are typically long, the system is prepared to offer the participants some rest, and it can be stopped if the user desires to do so at any moment. Thus, these cycles appear because, after the break, the dialogue is restarted from the beginning of the phase in which the conversation is stopped, keeping track of all the previously discussed topics and decisions.

Finally, we would also like to mention the white node of the figure: the *retrieve recipe* node. It is not coloured because it does not belong to any dialogue phase intrinsically. It is the recipe provider mentioned in Section 4.1 that is triggered when the user repeats the name of a given food. As expected, these kinds of system turns are triggered mostly in the Nutrition Introduction phase, where the participant tells the system about their nutrition routine.

In summary, the dialogue flow validates the design of our dialogue engine and also of the system, in general. This dialogue flow clearly corresponds to a successful implementation of the dialogue trees presented in Section 4.

### 8.4. Task-Completion

Task-oriented dialogue systems’ performance is usually measured via task-completion metrics. For instance, in the case of restaurant reservation, the task completion would indicate the percentage of dialogues where the system successfully books a restaurant satisfying the user constraints. Similarly but with a higher level of complexity, we analyse the percentage of the dialogue phases successfully finished throughout the dialogue with end-users. This is shown in Figure 13.

First of all, the task-completion is higher in Norwegian and French than in Spanish due to the aforementioned improvements in the DM for those two languages which mainly addressed problems that arouse in the later part of the dialogues. This indicates the importance of this module in the whole system.

We would like to remark the task-completion percentage at the Goal phase. It is 100% in Norwegian, very close to that number in French, and almost 80% in Spanish. This is already a very successful result, since establishing a goal the user would like to accomplish is the longest and most complex task in these dialogues. In fact, in the WoZ experiments, the dialogues where a user goal were found were considered successful [50]. However, due to the length and complexity of this first stage, many users were tired at this point. Hence, the principal drop in the task-completion is found between Goal and Motivation. In some cases, the session could not go further than the Goal phase because the users considered there was no need to change their nutrition habits. The subsequent drops are mostly caused by the system not being able to fulfill the objectives of each phase.

In the end, around 65% of participants in Norway and French were able to establish not only a goal but also a plan, while only around 25% of participants in Spain were able to do so. This suggests that our proposal is valid to produce long and complex dialogues which can potentially improve nutrition-related habits, especially after the improvements of the DM implemented for Norwegian and French.

### 8.5. NLG Performance

We measure the proposed NLG methodology in two different ways. In an offline and preliminary evaluation (Section 8.5.1), we compare two methodologies for the LM postprocessing. Then, in Section 8.5.2, we assess the performance in the VC.

#### 8.5.1. Offline Performance

As detailed in Section 6.5, there are cases where, given a set of candidate templates and attributes, GROWsetta has to select the combination that produces the grammatically and semantically correct sentence. In order to select the best output sentence, we run a language model over the sentences generated with all the possible combinations and select the one with the least perplexity/greater probability. We compared two proposals: an N-gram LM and a neural LM. While the N-gram model is a 3-gram model, for the neural LM, we trained a GPT-2 model for the three target languages from scratch, as described in Section 6.5). We show two results of the GPT-2 models for each language: after the first and second epochs of training.

We defined a set of tasks to analyse the capacity of the neural LM to select the correct candidates. A task consists of several tuples of sentences, where only one sentence per tuple is correct. The number of candidates in the tuples differs across the tasks. Table A1 of Appendix B shows all the tasks. These include, among others, cases where the verb tense has to match the number of the subject, or cases where a determinant that fits with the attribute has to be included. In Table 9, we present a sample task for each language.

Results per each task are shown in Table A2 of Appendix B. We summarise these results in Table 10, where the average accuracy at selecting the correct candidate per language is presented. The results demonstrate that the GPT-2 models clearly outperform the N-gram model in all the languages (and also in all the tasks), validating the inclusion of this kind of transformer models in the NLG. We would like to note that, whilst Spanish and French models improved the results after a second epoch of training, this did not happen in Norwegian, where the 1-epoch GPT-2 model was the best-performing LM. This can be caused by the less amount of good quality data available in that language, which may have caused the neural network to slightly overfit.

#### 8.5.2. Performance in the VC prototype

We also evaluated the performance of the NLG in the study with end users. As in [29], we have computed the ratio of user turns labelled as a repetition request by the NLU as a measure of the quality of the NLG: the more repetition request from the users, the more likely the NLG is producing not-understandable sentences. Note that this metric is a lower bound of the actual NLG performance; it may happen that the user does not understand the system due to inaccuracies in other modules, such as the TTS. The repetition request ratios obtained are shown in Table 11. Since the average number of turns per dialogue is 32.9, there is roughly only one repetition request per dialogue, which points out that the sentences produced by the NLG are highly comprehensible.

### 8.6. Human Acceptance

Finally, we measure participants’ perception of the VC prototype as well as their perception of the conversation flow, using an extended version of the VAAQ [85]. In addition to the four subquestionnaires of the VAAQ, a short questionnaire about the agent’s intelligibility was included too. In short, the users were asked about their perception of the system about the following qualities:**VAAQ Pragmatic qualities:** focus on the usefulness, usability, and accomplishment of the tasks of the proposed system, in this case, the GROW session;**VAAQ Hedonic qualities (identity):** related to the system’s personality;**VAAQ Hedonic qualities (feelings):** evaluate how captivating the system is, and how the users felt while conversing with it;**VAAQ Attractiveness:** assesses how tempting and attractive the interaction with the agent is;**Intelligibility:** evaluates the system’s output (generated language and voice).

The questions are formulated to be answered on a 5-point Likert scale, which allows computing the score of each subquestionnaire easily, between 0 and 100 in this case. The results, divided by language, are shown in Table 12. For comparison purposes, we also show the results of these questionnaires for the WoZ experiments, and highlight in bold the best results for each comparison.

According to Table 12, the system obtains mostly positive results (>50), which indicates the correct behaviour of the integrated VC and confirms the good design of the DM and NLG in this very challenging task. However, due to the complexity of developing automatic GROW sessions, there is still room for improvement, as shown by the difference between the automatic VC and WoZ results.

If we compare the results obtained in the three countries, the human perception of the Spanish system is similar to the French one, and better than the Norwegian one. This correlates well with conclusions extracted from turn lengths (in Section 8.2), and once again emphasises the influence of other modules besides the DM in SDSs, which are probably the cause of these differences, as previously explained. On the other hand, and as expected, VAAQ scores are higher for WoZ experiments than for the automatic system. Nonetheless, the differences are significant only in three cases, as opposed to the previous comparison in terms of turn length (see Figure 7). This indicates that, even if the WoZ system is notoriously more engaging and makes the users more talkative, their perception of the VC prototype is not significantly worse in many aspects.

To provide a more detailed view of the user’s perception of the system, we also show the score corresponding to seven specific questions related to the NLG and DM modules, in Table 13. These questions can help us gain a deeper insight into the positive points of the system, and also into its drawbacks. Questions marked with a dagger (^†^) ask about potential negative opinions on the system, but higher scores always mean higher performance.

The participants considered that communicating with the agent was rather simple and easy, and also not useless. In other words, the users were, in general, able to take advantage of the virtual GROW sessions. In comparison with the WoZ system, the biggest difference happens with the Norwegian system, due to the aforementioned reasons.

We also can conclude that the system is not very human, and the communication, even though useful and enjoyable (in Spanish and French), is not particularly engaging. This indicates that the interaction with the system is far from perfect, but since our work represents one of the first steps in building complex coaching systems, we find it acceptable. It is noteworthy that users find the VC prototype slightly more enjoyable than the WoZ system in Spanish and French. We hypothesise that this might be due to the increased delay of the WoZ system, produced by the wizard having to think and (sometimes) write the next response. Regarding how engaging the automatic VC is, the difference with the WoZ is once again notable (as in Section 8.2). This also suggests that using the turn length as an engagement metric can be appropriate.

When the participants were asked whether the communication was stressful, the French and Spanish answered quite strongly that it is not—even less than the WoZ system. According to the rest of our analysis, Norwegian users found it more stressful. Finally, the last question confirms the good performance of the NLG: the users in the three countries thought that the agent can easily be understood, which could not be possible if the NLG had produced grammatically or semantically incorrect sentences.

## 9. Conclusions

In this work, we have addressed the challenge to build a SDS with coaching strategies and communication skills to simulate the behaviour of health and coach professionals assisting independent elderly. To this end, we have specifically proposed a dialogue management strategy as well as a coaching language for robust interactions with the target community living in three very different countries, namely Spain, France and Norway.

The system has been successfully tested and validated in a study that incorporates human participants of the target community. The evaluation concludes that the proposed VC acts as expected and is capable to hold long and complex conversations in three different languages and cultures. Additionally, we would like to highlight the robustness of the system, which has been obtained due to the joint work of the core modules of the VC, the DM and NLG. However, a disadvantage of our dialogue engine is that it is not easily reproducible or scalable to other tasks. In terms of scalability to other languages, the DM does not require further work because it is language-independent. Nonetheless, the NLG and some additional modules of the system, such as the ASR or NLU, can be easily integrated following the same approach, but they do need the development of language-specific models.

On the other hand, the analysis of the participant’s perception of the prototype allows us to conclude that the communication with the VC is perceived as simple, easy, useful and enjoyable. Nevertheless, in a comparison with a WoZ system designed to carry out the same task, participants felt that our VC was significantly less engaging and human. Thus, future works may need to focus on addressing these weaknesses. Still, the VC was more engaging than a fully data-driven chatbot trained on the WoZ data. This highlights the current limitations of data-driven approaches and ratifies our choices for the DM and NLG methodologies. Other potential improvements would be to optimise the latency in the system response, which would lead to more natural interactions and better user experience. This could be achieved, for example, by integrating the system’s external services into edge devices.

Lastly, it would also be interesting to analyse the potential behaviour changes provoked after several coaching sessions in the future, and not only to evaluate the perception of the conversations. In fact, our dialogue engine is the basis of the SDS of the project GO-ON [86]. This SDS will be used in a clinical study about dementia to assess whether such systems (together with other tools) could be helpful to delay or prevent Alzheimer’s disease in the long term. 

## Figures and Tables

**Figure 1 sensors-23-01423-f001:**
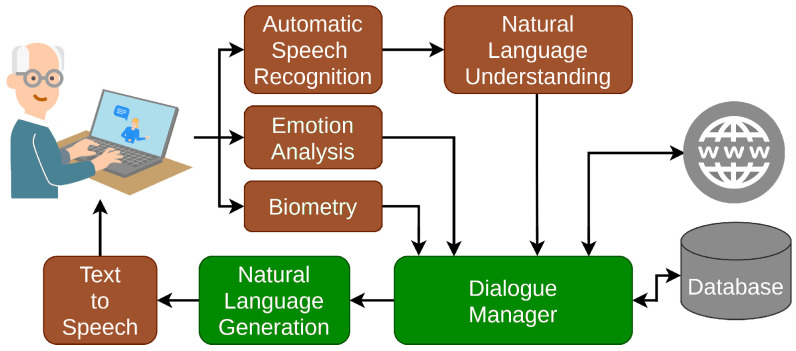
EMPATHIC system schema. The components we analyse here are highlighted in green.

**Figure 2 sensors-23-01423-f002:**
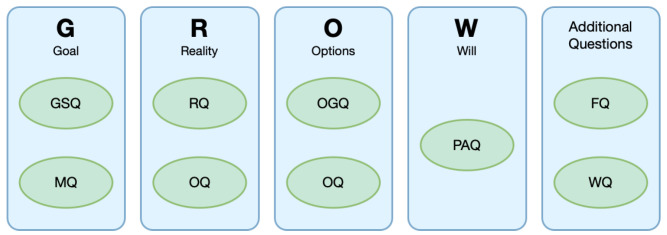
The structure of the GROW model of behaviour change [50].

**Figure 3 sensors-23-01423-f003:**
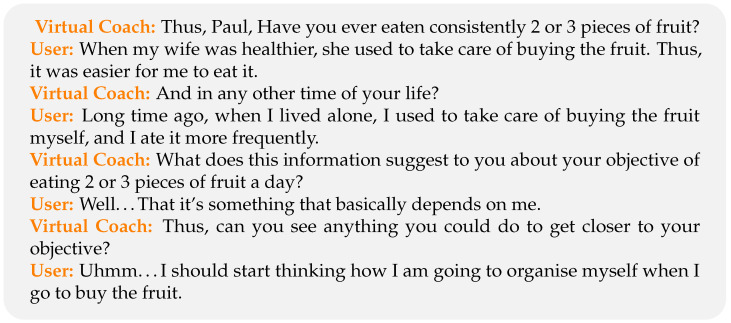
Example of a dialogue proposed by the Professional Coach.

**Figure 4 sensors-23-01423-f004:**
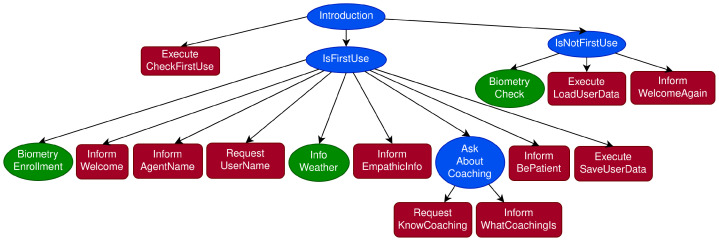
Task specification tree for the introductory dialogue. Internal agents or non-terminal nodes, which encapsulate a sequence of subtasks, are represented by blue nodes. Agents connecting to other modules in Figure 1 are represented by green nodes. Red nodes stand for terminal nodes that implement precise steps such as inform, request or execute.

**Figure 5 sensors-23-01423-f005:**
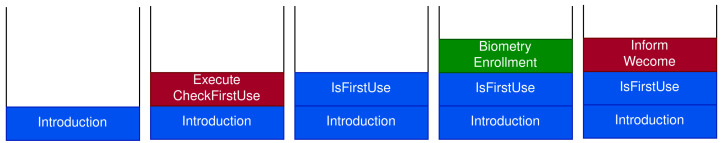
Movements of the stack when the DM traverses the specification tree in Figure 4 in Depth First order, and it considers the trigger values stored in the dashboard, as, for example, a true value as the output of the Execute CheckFirstUse terminal node.

**Figure 6 sensors-23-01423-f006:**
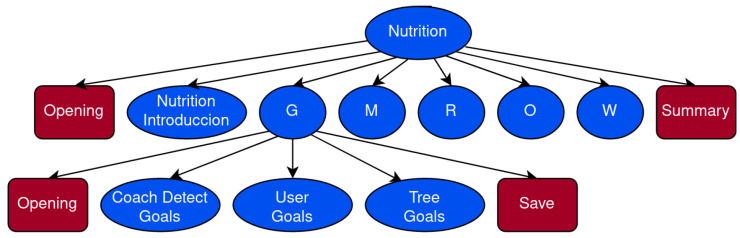
Task specification tree for the Nutrition agent. The sub-agent for the Goal (G) phase has been expanded.

**Figure 7 sensors-23-01423-f007:**
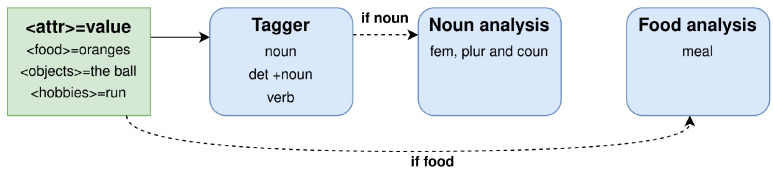
Entity analysis for GROWsetta.

**Figure 8 sensors-23-01423-f008:**
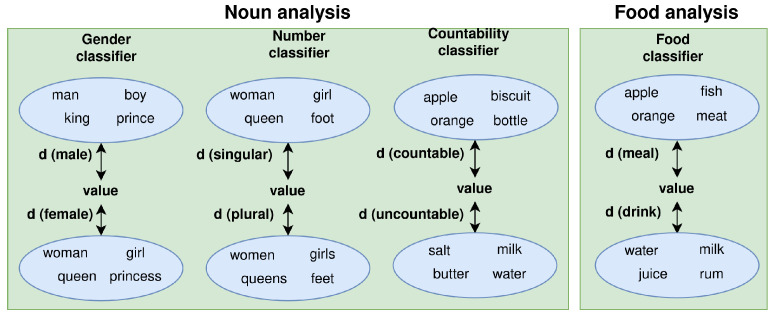
Noun and Food analysis for the GROWsetta process.

**Figure 9 sensors-23-01423-f009:**
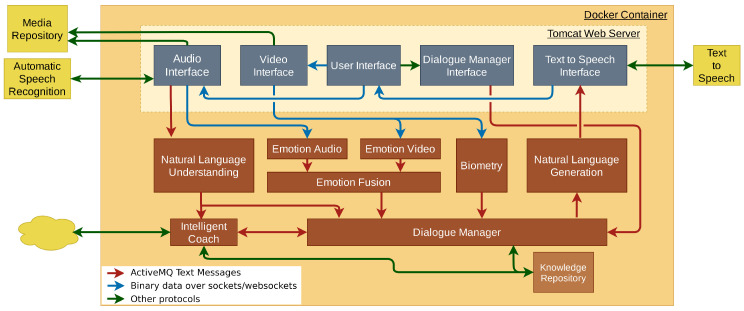
EMPATHIC system architecture. Orange is the Docker container. Brown components are running directly on the Docker container as well as the light yellow Tomcat Web server. Then, dark blue components are running as web services within the Tomcat Server. Yellow components are external services. Arrow colours indicate the communication protocol between the involved components.

**Figure 10 sensors-23-01423-f010:**
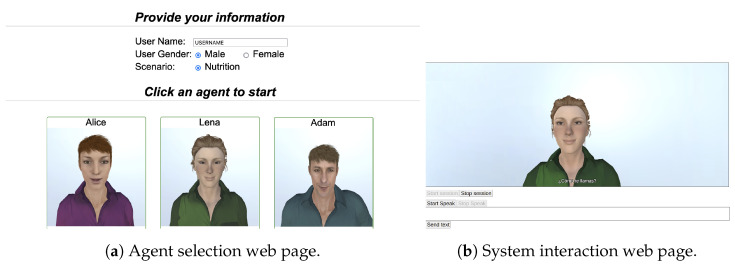
User Interface.

**Figure 11 sensors-23-01423-f011:**
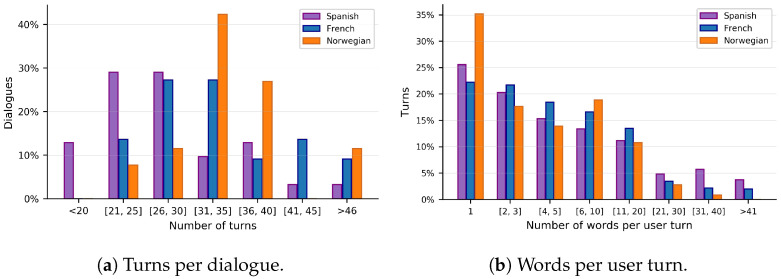
Histograms of the number of turns per dialogue and the number of words per user turn in the human evaluation of the final prototype.

**Figure 12 sensors-23-01423-f012:**
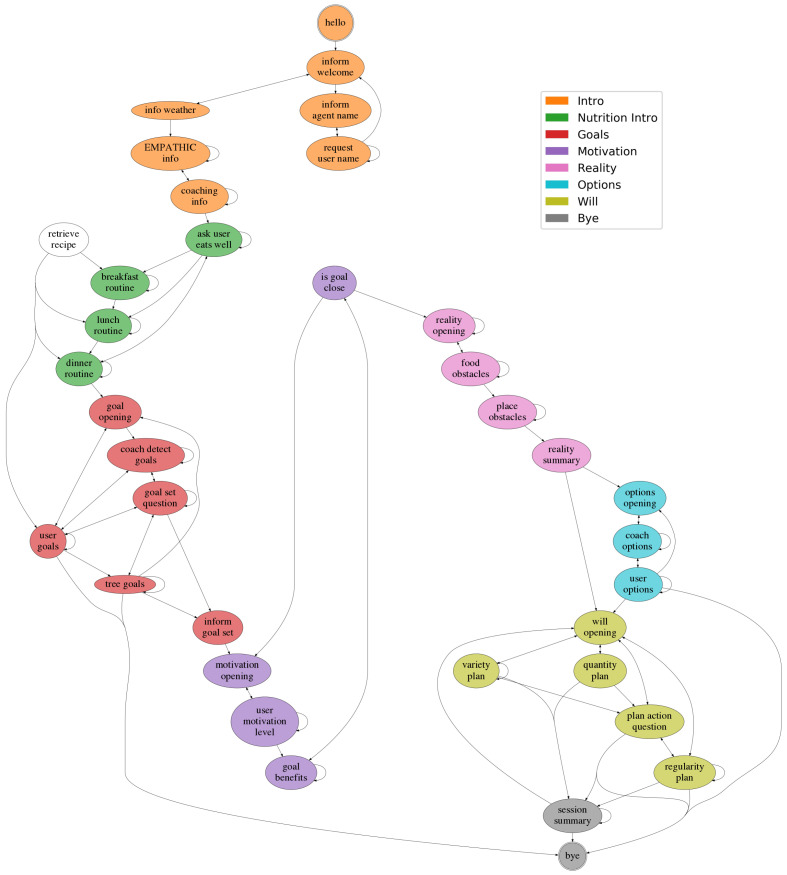
Dialogue flow graph obtained from the interactions between users and the final prototype of the VC. The nodes are groups of system turns and the arrows indicate common transitions. The colours indicate the dialogue phase the grouped turns belong to.

**Figure 13 sensors-23-01423-f013:**
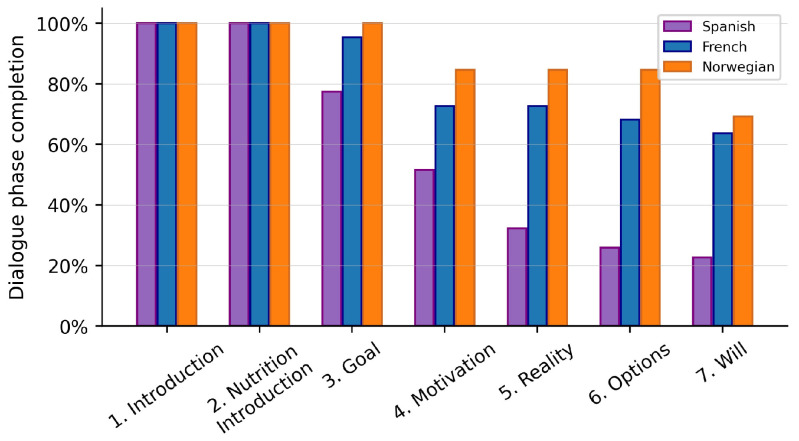
Percentage of dialogue phases the system successfully completed on average, per language.

**Table 1 sensors-23-01423-t001:** Extracts of a conversation between an end-user and the VC, which has been translated from Spanish into English. Column 2 depicts the DAs of each Coach turn, which defines the output of the DM and consistently the input of the NLG. Each DA generates a sentence shown in Column 3 that is part of the Coach turn under consideration. User turns only show the transcription of what the user said as well as the entities detected by the NLU, if so.

Turn	Dialogue Act	Sentence
**START OF THE CONVERSATION**		
**Coach**	Gen&Hello()	Hello!
	Gen&Inform(<<biometry>>=no_user)	I need you to locate in front of the camera to create your profile. If not, I can’t go on.
	Gen&Inform(<<biometry>>=accepted_first)	I have already created your profile.
	Int&self_introduction(<agent_name>=Natalie)	My name is Natalie. Nice to meet you.
	Int&what_name()	What is your name?
**User**	Juanjo NLU detected entities: {propernouns:[Juanjo]}	
**Coach**	Int&what_name(<<confirm_name>>=True, <user_name>=Juanjo)	Thus, your name is Juanjo. Is this correct?
**User**	yes NLU detected entities:{}	
**MID-CONVERSATION (GROW IN PROGRESS)**		
**Coach**	Gen&Pos_opinion()	Great!
	Gen&Open_topic(<<eating_routine_in_progress>>=False, <<eat_well>>=True)	It seems that you eat healthy but I would like to know the details of your nutrition routine to see if it can be improved.
	Gen&Open_topic(<topic>=breakfast)	Let’s focus on your breakfast.
	RQ&curr_sit(<<topic>>=breakfast)	Tell me what do you usually have for breakfast.
**User**	my usual breakfast consists of an apple and ....then a toast with oil NLU detected entities: {food:[apple, toast, oil]}	
**Coach**	RQ&curr_sit(<<echo>>=True,<<positive>=True, <action>=eat,<food>=apple_toast_oil)	Thus, you eat apples and toast and you drink oil.
	RQ&curr_sit(<<first>>=True,<<topic>>=breakfast, <food>=apple_toast_oil)	Do you think that having apples, toast and oil for breakfast gives you energy for all day?
**User**	I think so NLU detected entities: {}	
**END OF THE CONVERSATION**		
**Coach**	RQ&curr_sit(<<echo>>=True,<days_per_week>=2, <user_name>=Juanjo)	Juanjo, two per week?
	Gen&Pos_feedback(<<confirm>>=True)	Is this correct?
**User**	yes NLU detected entities: {}	
**Coach**	Gen&Pos_opinion()	Perfect!
	PAQ&what_plan(<<regular_time>>=True)	It’s important to define regular timetables to improve your regularity. Will you do it?
**User**	yes NLU detected entities: {}	
**Coach**	Gen&Pos_opinion()	I like that.
	Gen&Close_topic(<topic>=nutrition	Nutrition session has finished.
	GSQ&what_obj(<<summary>>=True,<action>=eat_regular)	The summary of the session is as follows. You have decided to improve your regularity.
	PAQ&what_plan(<<hours>>=True,<<action>>=eat_regular, <freq>=2,<food>=fish)	In addition, your first step to improve it is to eat fish twice a week defining a regular timetables for all the meals.
	Gen&Goodbye(<user_name>=Juanjo)	See you, Juanjo!

**Table 2 sensors-23-01423-t002:** The importance of “gender” (“Male” and “Female”) attribute in the three EMPATHIC languages addressed by GROWsetta.

Language	Agent Female User Male	Agent Female User Female	Agent Male User Male	Agent Male User Female
Spanish	Estoy cansada y veo que tú también estás cansado.	Estoy cansada y veo que tú también estás cansada.	Estoy cansado y veo que tú también estás cansado.	Estoy cansado y veo que tú también estás cansada.
Norwegian	Jeg er sliten og jeg ser at du er sliten.			
French	Je suis fatiguée et je vois que tu es fatigué.	Je suis fatiguée et je vois que tu es fatiguée.	Je suis fatigué et je vois que tu es fatigué.	Je suis fatigué et je vois que tu es fatiguée.

**Table 3 sensors-23-01423-t003:** Example of grammatical variation of the attribute value for the language-independent DA “enjoy_leisure”.

<action>=enjoy_leisure			
**Version**	**Spanish**	**French**	**Norwegian**
Infinitive	disfrutar del tiempo libre	profiter du temps libre	nyte fritiden
Present simple	aprovechas tu tiempo libre	vous vous amusez pendant votre temps libre	nyter fritiden
Antonym infinitive	aburrirte en tu tiempo libre	ne pas profiter du temps libre	ikke nyte fritiden
Antonym present simple	te aburres en el tiempo libre	vous ne profitez pas du temps libre	nyter ikke fritiden

**Table 4 sensors-23-01423-t004:** Examples of sentences selected via Transformer LM.

Value Attribute	Templates	Generated Sentences	Correct
<dates>=yesterday	What are you going to do <dates>?	What are you going to do yesterday?	No
	What did you do <dates>?	What did you do yesterday?	Yes
<dates>=Mondays	What are you going to do <dates>?	What are you going to do Mondays?	No
	What are you going to do on <dates>?	What are you going to do on Mondays?	Yes

**Table 5 sensors-23-01423-t005:** Statistics of the corpora used to pretrain the GPT-2 model in Spanish, French and Norwegian. In Norwegian, values in brackets refer to the data prior to the addition of the OSCAR corpus.

	Spanish	French	Norwegian
Amount of raw text	10 GB	7 GB	5 GB (1 GB)
Number of sentences	230 M	121 M	30 M (14 M)
Running words	1.7 B	1.3 B	750 M (150 M)

**Table 6 sensors-23-01423-t006:** Information about the participants in the human evaluation.

	Spain	France	Norway
Nb. of participants	31	22	26
Avg. age	71.6	68.4	73.4
Female participants	44.8%	53.3%	30.8%
Avg. quality of life	68.6	65.1	75.2
Avg. GDS depression level	4.2	6.8	3.7
Avg. ease of use of PCs	82.7	79.4	93.3

**Table 7 sensors-23-01423-t007:** Average number of words per user turn in WoZ, automatic VC, and coaching chatbot tests. The mark * indicates statistical significance, and bold values indicate the best results.

	WoZ Experiments	VC Prototype	Data-Driven Bot
Spanish	**12.9** *	9.5	3.6 *
French	**18.2** *	7.6	3.4 *
Norwegian	**17.9** *	5.4	5.3

**Table 8 sensors-23-01423-t008:** Average response time and standard deviation per language, in seconds.

Spanish	French	Norwegian
4.23 ± 0.80	3.77 ± 0.98	4.82 ± 1.21

**Table 9 sensors-23-01423-t009:** Examples of tasks to analyse the LMs’ performance at selecting the correct template given an attribute. In the examples, the attribute is underlined. All tasks are in Table A1 of Appendix B.

Name	Nb. of Tuples	Nb. of Options	Brief Description	Correct Sentence Example	Incorrect Sentence Example
es_verb_time	1000	2	The verb has to match the adverbial of time.	¿Y qué ha sucedido ayer?	¿Y qué sucederá ayer?
fr_verb_num	120	2	The verb conjugation has to match the number of the subject.	Que vous ont apporté les vins?	Que vous a apporté les vins?
no_verb_prep	104	4	The attribute has to fit with the verb and the preposition. Its placement has to be correct as well.	Ønsker du å spise nå?	Ønsket du å spise i nå?

**Table 10 sensors-23-01423-t010:** Template selection average accuracies per language. The models are different for each language and also the number of candidates per task. Results for all tasks can be found in Table A2 of Appendix B. Bold values indicate the best results for each language.

Accuracy	N-Grams (N = 3)	GPT-2 1 Epoch	GPT-2 2 Epochs
es	38.85	52.23	**84.35**
fr	39.75	49.68	**62.39**
no	26.47	**76.94**	73.86

**Table 11 sensors-23-01423-t011:** NLG errors measured as the repetition request ratio.

Spanish	French	Norwegian
3.4%	3.5%	5.0%

**Table 12 sensors-23-01423-t012:** VAAQ average score per subquestionnaire for WoZ experiments, and for the VC prototype. The mark * indicates statistical significance.

WoZ/VC Proto.	Pragmatic Qualities	Hedonic Qualities (Identity)	Hedonic Qualities (Feelings)	Attractiveness	Intelligibility
Spanish	**63.0**/58.5	**71.7** */65.4	**62.5** */52.5	**64.7**/61.4	**71.0**/63.6
French	**60.8**/51.8	**77.0**/71.4	**64.4** */45.7	**66.9**/61.8	62.5/**67.8**
Norwegian	**57.2**/47.5	**70.9**/67.0	**56.9**/48.9	**57.3**/50.8	**64.9**/63.6

**Table 13 sensors-23-01423-t013:** Scores of seven VAAQ questions for WoZ experiments, and for the VC prototype. Values in bold indicate the best result for each comparison, and the mark * shows if the difference is statistically significcant.

WoZ/VC Proto.	Spanish	French	Norwegian
I think that communicating with the agent is simple and easy.	**72.3**/66.7	53.1/**57.6**	**66.0**/53.8
I think that communicating with the agent is useless. ^†^	**70.0**/66.7	**67.9**/62.0	**63.3**/54.8
I think the agent is very human.	**48.7** */34.2	**57.7** */40.2	**48.9**/46.2
I think that communicating with the agent is enjoyable.	54.7/**63.3**	60.3/**63.0**	**45.1**/38.5
I think that communicating with the agent is engaging.	**69.6** */56.9	**72.3** */48.9	**60.1**/52.9
I think that communicating with the agent is stressful. ^†^	76.0/**78.3**	85.3/**87.0**	**67.1**/55.8
The agent can be easily understood.	**88.0**/82.5	75.0/**82.6**	82.3/**82.7**

## Data Availability

The Empathic data corpus is distributed for research purposes by the European Language Resources Association (ELRA) http://www.elra.info/en/about/, accessed on 20 January 2023 at a very low price for academic and research institutions, as well as for small companies. ELRA catalogue can be found here http://catalogue.elra.info/en-us/, accessed on 20 January 2023. Corpus ISLRN: 631-345-309-445-9 and ELRA ID: ELRA-S0414.

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
