# Peer review of "Dialogue Management and Language Generation for a Robust Conversational Virtual Coach: Validation and User Study"

_sensors, 2023, doi:10.3390/s23031423_

Round 1
Reviewer 1 Report
The paper proposed a dialogue management and language generation-based virtual coach application. The natural language generation combines the classical template-based approach with transformer-based LM, where the SOTA GPT-2 is adopted as a pre-trained model.
The paper is organized well, where the details of design, implementation, and experiment are solid.
Based on the present form, the manuscript is supposed to give results of response time of conversation in experiments. It will perform the effectiveness and efficiency of the proposed virtual coach in real-time application scenario.
Reviewer 2 Report
Is this virtual coach is restricted to those three regions? Or is it generic enough to be used across the world?
The VC system is running in cloud (as per my understanding). Is it not worthwhile to use edge devices towards continuous learning? If the cloud is situated at a faraway place, the real-time interaction is possible?
